# Identification and Fine Mapping of a Locus Related to Leaf Up-Curling Trait (Bnuc3) in *Brassica napus*

**DOI:** 10.3390/ijms222111693

**Published:** 2021-10-28

**Authors:** Shubei Wan, Zongping Qin, Xiaomei Jiang, Mao Yang, Wenjing Chen, Yangming Wang, Fei Ni, Yijian Guan, Rongzhan Guan

**Affiliations:** 1State Key Laboratory for Crop Genetics and Germplasm Enhancement, Nanjing Agricultural University, Nanjing 210095, China; 2016201079@njau.edu.cn (S.W.); 2019101078@njau.edu.cn (Z.Q.); 2019101077@njau.edu.cn (X.J.); 2017201033@njau.edu.cn (M.Y.); 2016201058@njau.edu.cn (W.C.); 2015201029@njau.edu.cn (Y.W.); 2019201051@njau.edu.cn (F.N.); rapeseed@njau.edu.cn (Y.G.); 2Jiangsu Collaborative Innovation Center for Modern Crop Production, Nanjing Agricultural University, Nanjing 210095, China

**Keywords:** up-curling leaf, mapping, *Brassica napus* L.

## Abstract

Leaf trait is an important target trait in crop breeding programs. Moderate leaf curling may be a help for improving crop yield by minimizing the shadowing by leaves. Mining locus for leaf curling trait is of significance for plant genetics and breeding researches. The present study identified a novel rapeseed accession with up-curling leaf, analyzed the up-curling leaf trait inheritance, and fine mapped the locus for up-curling leaf property (*Bnuc3)* in *Brassica napus*. Genetic analysis revealed that the up-curling leaf trait is controlled by a single dominant locus, named *BnUC3*. We performed an association study of *BnUC3* with single nucleotide polymorphism (SNP) markers using a backcross population derived from the homozygous up-curling leaf line NJAU-M1295 and the canola variety ‘zhongshuang11’ with typical flat leaves, and mapped the *BnUC3* locus in a 1.92 Mb interval of chromosome A02 of *B. napus*. To further map *BnUC3*, 232 simple sequence repeat (SSR) primers and four pairs of Insertion/Deletion (InDel) primers were developed for the mapping interval. Among them, five SSR markers and two InDel markers were polymorphic. By these markers, the mapping interval was narrowed to 92.0 kb using another F_2_ population. This fine mapping interval has 11 annotated genes among which *BnaA02T0157000ZS* were inferred to be candidate casual genes for up-curling leaf based on the cloned sequence analysis, gene functionality, and gene expression analysis. The current study laid a foundational basis for further elucidating the mechanism of *BnUC3* and breeding of variety with up-curling leaf.

## 1. Introduction

In plants, the leaf, the primary photosynthetic organ, is vital for growth and development. Several leaf traits, including size and rolling, significantly influence plant production. Moderate leaf up-curling improves photosynthetic efficiency and increases yield. It reduces the incident solar radiations and leaf transpiration and subsequently provides drought resistance [1]. Therefore, mining the genes associated with leaf up-curling will uncover the underlying molecular mechanisms and help plant breeding. 

In higher plants, few mechanisms leading to leaf curling have been discovered. The leaf-curling genes have been associated with leaf polarity establishment, cutin and cuticular wax biosynthesis, cell wall synthesis, and hormone responsiveness. In particular, leaf adaxial–abaxial polarity has been crucial for forming curled leaves [2,3]. *PHANTASTICA* (*PHAN*), encoding an MYB domain transcription factor, was first associated with adaxial-abaxial polarity in *Antirrhinum majus* [4,5]. In *Arabidopsis*, *ASYMMETRIC LEAVES 1* (*AS1*), a homolog of *PHAN*, regulates leaf adaxial polarity, forming a protein complex with the plant-specific lateral organ boundaries (LOB) family protein AS2 [6]. Another three members of the *HD-ZIP III* (Class III homeodomain-leucine zipper) gene family, including *PHABULSA* (*PHB*), *PHAVOLUTA* (*PHV*), and *REVOLUTA* (*REV*), have also been associated with leaf-rolling phenotype in *Arabidopsis* [7,8]. The loss-of-function mutants of *HD-ZIP III* genes in maize, rice, and cucumber had adaxial cells and rolled leaves [9,10,11,12]. Besides, *KANADI* (*KAN*) genes, the members of the GARP (Golden2, ARR-B, and Psr1) family of transcription factors, regulate leaf abaxial polarity establishment by inhibiting the expression of *HD-ZIP III* and *AS2* in the abaxial of leaves [13,14,15,16]. *KAN* regulates the expression of downstream genes, such as *YABBY*, which encodes YABBY domain proteins regulating leaf abaxial patterning [17,18,19]. AUXIN RESPONSE FACTORs (ARFs), including ETTIN (ETT, also known as ARF3), ARF4, and ARF2, form a complex with KAN implicated in abaxial polarity establishment [20,21,22]. Furthermore, few non-coding RNAs have also been found to be involved in establishing adaxial-abaxial polarity. The microRNAs miR165 and miR166 are known to target *HD-ZIP III* [16,23], while miR390 triggers the production of phasiRNAs from TAS3 ta-siRNA (trans-acting short interfering RNAs) transcripts, repressing the activity of ARFs [24,25,26,27]. Thus, various genes and non-coding RNAs affect cell wall biogenesis as leaf polarity signals, leading to mechanical heterogeneity of the cell wall [28]. 

Studies have also associated the genes shaping leaves with the process of leaf cell wall formation and maturation. In rice, *RL14* (*rolling-leaf14*) encoding a 2OG-Fe (II) oxygenase family protein affects leaf water transport by altering secondary cell wall composition. Loss-of-function mutation in *RL14* resulted in leaf incurving by the shrinkage of bulliform cells on the adaxial side [29]. Moreover, cuticle wax is considered a critical factor affecting leaf shape. Eight *Arabidopsis* T-DNA insertion mutants, defective in cuticular wax biosynthesis, showed leaf-curling phenotype due to loosely arranged epidermal cells [30]. In maize, a mutation in the gene *Zmsrl5* (*semi-rolled leaf 5*) related to cuticular wax biosynthesis changed leaf epidermal bulliform cells, resulting in semi-rolled phenotype [31]. Meanwhile, the transgenic *Arabidopsis* overexpressing *SHINE*, encoding an APETAL2 domain transcription factor that regulates wax biosynthesis, had up-curling leaves due to increased cuticular wax and curved-down edges [32]. 

Furthermore, phytohormones such as auxin and brassinosteroids are related to the formation of curled leaves [33]. The loss-of-function mutation in *AUXIN RESISTANT6* (*AXR6*), encoding *CULLIN1*, a subunit of SCF (Skp1–Cullin-1–F-box) complex, showed reduced auxin response gene expression and rolled-leaf phenotype [34]. In *Arabidopsis*, the *ULTRACURVATA1* (*UCU1*) gene, encoding an AtSK (formerly designated ASK) protein, is involved in the cross-talk between the auxin and brassinosteroid signaling pathway; leaves of the mutant *ucu1* were rolled spirally downward [35]. Meanwhile, the rolled-leaf phenotype of *Arabidopsis BRASSINOSTEROID-INSENSITIVE 1* (*BRI1*) mutant (*bri1*) indicated the role of *BRI1* in leaf shaping [36,37].

Germplasm with up-curling leaves have been used in rice and maize genetic improvement for elevating cereal crops yield [31]. To find the method to improve canola variety yield, our previous studies have mapped two loci associated with up leaf curling (*BnUC1* and *BnUC2)* onto chromosome A05 of rapeseed (*Brassica napus* L.) [2,38], and our present study reports a new *B. napus* line NJAU-M1295 with up-curling leaves (*Bnuc3*). With this novel accession, we fine mapped the locus responsible for the up-curling trait by single nucleotide polymorphism (SNP) and other molecular markers, and identified the candidate gene for the trait based on gene sequencing and expression analyses. Further, we evaluated the effects of the locus on the photosynthetic properties. Our findings may help elucidate the mechanisms underlying the leaf trait and offer clues for breeding.

## 2. Results

### 2.1. The Leaf Up-Curling Mutant Performance 

The NJAU-M1295 plant with up-curling leaves was crossed with the elite variety Zhongshuang 11 (ZS11) with typical flat leaves. Leaves of the F_1_ plants were curled up, similar to NJAU-M1295 (Figure 1), indicating up-curling as a dominant trait. 

The leaf chlorophyll (Chl) content and photosynthetic indicators of the F_2_ population were determined to evaluate the effect of the up-curling of leaves. The leaf chlorophyll a (Chl a), chlorophyll b (Chl b), and total Chl levels (Table 1) were not different between the plants with up-curling leaves and those with normal, flat leaves. Measurement of the gas-exchange parameters leaves. Meanwhile, the stomatal conductance, intercellular CO_2_ concentration, and leaf transpiration rate of up-curling leaves were significantly higher than those of flat leaves (Table 2). These observations indicate the negative effect of leaf up-curling on photosynthetic efficiency to a certain extent.

### 2.2. Inheritance of the Up-Curling Leaf Trait

The NJAU-M1295 with up-curling leaves was crossed with two parents (ZS11 and NJAU-CP3756) with normal, flat leaves to produce F_1_, F_2_, and backcross (BC_1_) populations to clarify the genetic regulation of leaf shape. The leaf morphological traits of these populations were analyzed at seedling stage (Table 3). Data collected showed that the two F_2_ populations agreed with the expected Mendelian segregation ratio of 3:1, and two BC_1_ populations agreed with the expected Mendelian segregation ratio of 1:1. These results imply that the up-curling of leaves is a trait controlled by a single pair of dominant loci, referred to as *BnUC3* in this study.

### 2.3. Mapping of the BnUC3 Locus 

Thirty-seven individuals from the backcross population (ZS11× (ZS11×NJAU-M1295)) were genotyped using a Brassica 60 K SNP BeadChip Array (Illumina, San Diego, CA, USA) which has 52,157 SNP markers. Then, the association of *BnUC3* with SNPs was analyzed by genome-wide SNP scan using TASSEL 5 software [39]. In this analysis, the up-curling phenotype was given a value of 1, and the normal phenotype was given a value of 0. Results showed that the *BnUC3* locus was tightly associated with two SNP markers, M49061 and M02825, on the A02 chromosome (Figure 2). Chi-square test (χ^2^) was conducted to evaluate the association further (Table 4), indicating tight linkage between *BnUC3* and the two markers. Additionally, the SNP genotypes between and near M49061 and M02825 were checked, and one recombinant was found between M49061 and *BnUC3* and two between M02825 and *BnUC3* in the backcross population with ZS11 as the recurrent parent. These observations indicated that *BnUC3* was probably located in the 1922 kb long interval between M49061 and M02825 (Figure 3a). 

However, no polymorphic SNP markers were detected between M49061 and M02825 in the backcross population. Besides, simple sequence repeat (SSR) markers developed also exhibited no polymorphism. Therefore, another mapping population was used to map *BnUC3*. The F_2_ population, derived from NJAU-M1295 with up-curling leaves and NJAU-CP3756 with flat leaves, was used. Then, 232 pairs of SSR primers were developed to screen the markers linked to the *BnUC3* locus (see Appendix A) using the replaced population with 584 individuals, based on the genomic sequence of the preliminary mapping interval. Analysis of the F_2_ population revealed five polymorphic SSR markers (BnaA02V0056, BnaA02V0099, BnaA02V0333, BnaA02V0573, and BnaA02V0915). These markers covered a distance of 2.827 cM, corresponding to 1135.5 kb (Figure 3b,c). 

Subsequently, the linkage map generated with JoinMap showed that *BnUC3* was localized in the interval between BnaA02V0099 and BnaA02V0333, 0.182 cM from *BnUC3* to BnaA02V0099 (Figure 3b,c). No additional polymorphic SSR markers were found between BnaA02V0099 and BnaA02V0033. The renewed mapping interval genomic sequence of *B. napus* cv. ZS11 was compared with that of *B. napus* cv. Darmor (http://www.genoscope.cns.fr/brassicanapus/, accessed on 25 October 2021) to narrow down the mapping interval, revealing some insertion/deletion (InDel) sites. Based on their sequence variation, four pairs of primers were developed, and two polymorphic InDel markers, BnaA02INDEL1 and BnaA02INDEL2, were obtained (Figure 4). Marks scanned in the F_2_ population using these markers resulted in a narrowed mapping interval of 92.0 Kb (0.357 cM long), with *B. napus cv.* ZS11 genome as the reference (Figure 3d,e).

### 2.4. Candidate Gene A nalysis

The homologous segment sequences of the fine-mapped interval were downloaded from *B. napus cv.* ZS11 and Darmor, and the genes were annotated using *A. thaliana* annotation and ZS11 genome as the reference to identify the candidate genes associated with the leaf up-curling trait. The interval harbored 11 annotated genes (Table 5), and the genes *BnaA02T0156700ZS*, *BnaA02T0157000ZS*, and *BnaA02T0157100ZS* were listed as candidates. *BnaA02T0156700ZS*, homologous to *AT1G65440*, encodes a putative WG/GW-repeat protein (SPT6L) that regulates HD-ZIP III. Studies have associated this gene with the curly leaf formation in several species [12,40,41]. *BnaA02T0157000ZS*, homologous to *AT1G65510*, encodes a Brassicaceae-specific secreted transmembrane peptide (STMP). STMP overexpression led to small and curled leaves [42]. *BnaA02T0157100ZS*, homologous to *AT1G65520*, encodes an enoyl-CoA delta isomerase 1 (ECI1) that converts indole-3-butyric acid (IBA) to active indole-3-acetic acid (IAA) [43]. This gene affects the IAA biosynthesis pathway and may be associated with the leaf up-curling [44]. Meanwhile, other genes in the mapping interval cannot be completely excluded from the candidate genes list, although they have not been associated with up leaf curling.

Furthermore, the 11 genes’ coding sequences (CDS) were cloned and sequenced with gene-specific primer pairs from the two parents of the mapping population (NJAU-M1295 and NJAU-CP3756). Sequence alignments with ZS11 genome as the reference showed that gene coding sequence of *BnaA02T0157000ZS* has a nucleotide mutation in NJAU-M1295, causing a change in the functional product STMP at position 68 (Arg) into His in the plant with up-curling leaves (Figure 5). This mutation is in the disorder region as predicted by mobiDB-lite (https://www.ebi.ac.uk/interpro/, accessed on 25 October 2021), and it may have affected the peptide secretion function, and probably altered the leaf morphology. The other 10 genes in the mapping interval did not show sequence differences between NJAU-M1295 and NJAU-CP3756.

The expression of the *BnaA02T0157000ZS* was analyzed by qRT-PCR using leaf samples from the F_2_ subpopulation with up-curling leaves and the subpopulation with flat leaves. Analysis revealed significantly higher expression levels of *BnaA02T0157**000ZS* in the plants with up-curling leaves than plants with flat leaves (Figure 6). In summary, as revealed by the gene sequencing and potential gene functionality analysis, the STMP mutation may be inferred to be responsible for the dominant leaf-curling trait.

## 3. Discussion

Moderate leaf curling minimizes shadowing between leaves and improves crop yield by increasing planting density [45,46]. It also reduces leaf transpiration under drought stress [47]. Therefore, studies on leaf living states are of significance. In *Oryza sativa*, over 70 genes/QTLs for the rolled leaf trait have been reported, and more than 17 genes involved in leaf curling have been cloned [46,48]. Moreover, the last three leaves of a super, high-yielding hybrid rice were rolled (V-shaped) [49]. Leaf curling has also been utilized in maize breeding for compact plant types. Six mutants have been characterized in maize with leaf curling traits [31,50]. In soybean, mutants with rolled leaves and genes/QTL associated with rolled leaves have been reported [51]. Recent studies identified two loci related to rolled leaves (*BnUC1*, *BnUC2*) in rapeseed [2,38]. The present study identifies a novel leaf up-curling locus (*BnUC3*), which may help in breeding and genetic research on leaf living state in rapeseed.

In this study, limited plants were initially used for SNP genotyping to map and locate *BnUC3*. This approach was inexpensive and practical due to the relatively short genomic length corresponding to per cM of *B. napus* [2,49]. The SSR marker scan designed for the mapping interval compensated for the shortfall in SNP marker nonuniformity in the rapeseed genome and helped narrow down the mapping interval [52,53]. However, these markers were not enough for fine mapping due to a lack of polymorphic SSR markers. Therefore, the genomes of multiple rapeseed accessions were aligned that helped find the structural variation (SV). Further, based on the SVs, InDel markers were developed and used to narrow down the mapping interval. This study thus indicates that the jointed multiple modern marker detection approaches may help map the target locus. 

Chlorophyll content and photosynthetic efficiency may affect the yield of rapeseed [54]. In this work, the chlorophyll content did not change significantly between up-curling leaves and flat leaves; however, the photosynthetic efficiency of the plants with the up-curling leaves was lower than those with the flat leaves, possibly associated with the alteration in leaf adaxial–abaxial polarity (unpublished). This phenomenon has been observed in many other species with up-curling leaves [29]. However, it does not imply that up-curling leaf trait may not be used in breeding. Utility of up-curling trait plays roles in increase of planting density and in improvement of planting population light acceptation structure and air ventilation among plants, which are useful in elevating overall yield per unit area. 

Furthermore, the gene responsible for the up-curling trait showed a mutation in the conserved region of STMP specific to Brassicaceae. This gene had two copies, unlike other genes with at least three copies in the allotetraploid *B. napus*. Therefore, the dominant gene readily exhibited the mutant phenotype. Although the photosynthetic efficiency was lower in the mutant, the potential gene responsible for the up-curling of leaves was not found to affect photosynthesis. Therefore, studies should further explore the mechanisms underlying the candidate gene.

## 4. Materials and Methods

### 4.1. Plant Materials 

The *B. napus* line NJAU-M1295 with up-curling leaves from the germplasm bank of Nanjing Agricultural University was crossed with Zhongshuang11 (ZS11) with normal, flat leaves, and the F_1_ was selfed and backcrossed with ‘ZS11’ to generate the F_2_ and BC_1_ populations, respectively. NJAU-CP3756 with normal, flat-leaf phenotype was also used to create the mapping populations. 

### 4.2. Determination of Chlorophyll Content and Photosynthetic Efficiency

Chlorophyll (Chl) and carotenoids were extracted from 0.2 g of fresh leaves with 50 mL of 80% acetone, and the concentrations were measured using an Alpha-1500 spectrophotometer (LASPEC, Shanghai, China) [2]. The chlorophyll a (Chl a), chlorophyll b (Chl b), total Chl, and carotenoid concentrations were measured as described previously [55,56]. Each experiment had five biological replicates. The mean values of all the traits were compared between the up-curling leaf and the flat leaf plants by Student’s *t*-tests.

The photosynthetic parameters of the plants with up-curling leaves and the plants with flat leaves were determined at the rosette stage using Li-Cor 6400 portable photosynthesis system (Li-Cor Inc., Lincoln, NE, USA) as described by Zhang et al. (2010) [57]. Each experiment had four biological replicates. The mean values of all the traits were compared between the up-curling leaf and the flat leaf plants by Student’s *t*-tests.

### 4.3. Inheritance of the Leaf Curling Trait

Genetic segregation was investigated using F_2_ and backcross populations derived from the parent NJAU-M1295 with up-curling leaves and the parents with the flat leaves (ZS11, NJAU-CP3756). The χ^2^ test assessed the segregation ratios. 

### 4.4. SNP Genotyping and Locus Mapping

Total DNA was isolated from the fresh leaves using a modified cetyltrimethylammonium bromide (CTAB) method [58]. The extracted DNA was digested using RNase I (Takara, Dalian, China) to remove RNA, and the DNA concentration was adjusted to 200 ng µL^−1^. DNA samples from three plants with flat leaves and 34 plants with up-curling leaves in the BC1 population (NJAU-M1295×ZS11)×ZS11 were used for SNP genotyping, performed as described previously [59]. Association mapping was conducted to map the locus controlling the leaf curling trait using the SNP markers and leaf shape data, and was performed using software TASSEL 5 [39]. In this analysis, the up-curling phenotype was given a value of 1, and the normal phenotype was given a value of 0, and marker genotype was causal variable [60]. This approach is in fact a conventional linkage mapping method for a qualitative trait locus. Chi-square test (χ^2^) was conducted to evaluate the association between SNP markers (M49061 and M02825) and leaf up-curling trait.

After the initial mapping of *BnUC3* onto the A02 chromosome, sequence of the mapped interval on the A02 chromosome were downloaded from *B. napus* pan-genome information resource (BnPIR; http://cbi.hzau.edu.cn/bnapus/, accessed on 25 October 2021), and SSR markers were identified using SSR Hunter 1.3 program [61], and SSR marker primers were designed using Premier 5.0 [62]. The PCR for the SSR markers was performed as previously described [2]. A total of 232 primer pairs (see Appendix A) were designed based on the mapping interval genome sequence of ZS11 (http://cbi.hzau.edu.cn/bnapus/, accessed on 25 October 2021). However, no polymorphic SSR markers were found in the preliminary mapping interval generated by the SNP markers. Further, a canola line NJAU-CP3756 from our lab germplasm bank replaced ZS11 to generate the mapping population. The polymorphic SSR markers (BnaA02V0056, BnaA02V0099, BnaA02V0333, BnaA02V0573, and BnaA02V0915) were detected in the F_2_ population derived from up-curling leaf parent NJAU-M1295 and flat-leaf parent NJAU-CP3756. A total of 584 individuals in this F_2_ population were analyzed to generate a linkage map and determine the chromosome bearing the five SSR markers by JoinMap 4.1 software [63]. 

Further, the genomic sequences within the preliminary mapping interval were downloaded from *B. napus* Genome Browser (http://www.genoscope.cns.fr/brassicanapus/, accessed on 25 October 2021) and *B. napus* pan-genome information resource (BnPIR; http://cbi.hzau.edu.cn/bnapus/, accessed on 25 October 2021) for fine mapping the *BnUC3* locus. The sequences of the chromosomal segments obtained from the reference genomes were compared to identify the InDels. Based on this, four InDel marker primers (see Appendix A) were designed using Premier 5.0 [62], and two InDel markers (BnaA02INDEL1 and BnaA02INDEL2) were identified as polymorphic, which narrowed the mapping interval. PCR of InDel markers was performed in a 20-μL reaction volume using 2×EasyTaq^®^ PCR SuperMix (Transgen, Beijing, China). Standard PCR program was adopted as follows: an initial denaturation of 94 °C for 5 min, followed by 35 cycles of denaturation at 94 °C for 45 s, annealing at Tm (specific for each marker) for 45 s, and extension at 72 °C for 2 min, and a final extension at 72 °C for 5 min. Then, the polymorphic bands were separated in agarose gel (1%).

### 4.5. Candidate Gene Analysis 

Homologous sequence and annotation information of the mapped interval on the A02 chromosome were downloaded from *B. napus* Genome Browser (http://www.genoscope.cns.fr/brassicanapus/, accessed on 25 October 2021) and *B. napus* pan-genome information resource (BnPIR; http://cbi.hzau.edu.cn/bnapus/, accessed on 25 October 2021) to analyze the candidate gene.

### 4.6. Sequence Analysiss 

Genomic regions containing the CDS of the candidate genes were cloned from NJAU-M1295 and NJAU-CP3756. The specific primers (see Appendix A) were designed using Premier 5.0 [62]. The PCR program was adopted as described previously [64]. The amplified fragment was inserted into the pEASY-Blunt Cloning Kit vector (TransGen, Beijing, China) and sequenced. The resulting sequences were aligned using Clustal X software [65].

### 4.7. Quantitative RT-PCR Analysis

Total RNA was isolated from the leaves at the tillering stage using the RNAprep Pure Plant Kit (BioTeke, Beijing, China). The first-strand complementary DNA was synthesized using a reverse transcription kit (Takara, Tokyo, Japan). The specific primers (see Appendix A) of candidate genes detected in the mapped interval and reference gene *Actin7* were designed using Premier 5.0 [62]. The qRT-PCR was performed on a CFX96 Touch Real-Time PCR Detection System (BIO-RAD, Hercules, CA, USA) with SYBR Green Real-time PCR Master Mix (Toyobo, Tokyo, Japan). Three biological replicates were used. Relative expression levels were calculated using the 2^−ΔΔCt^ method [66]. The gene expression comparison between the up-curling leaf and the flat leaf plants was conducted by Student’s *t*-tests.

## 5. Conclusions

A new leaf mutant with an up-curling leaf, NJAU-M1295, was discovered from our *B. napus* germplasm. Inheritance studies showed that the up-curling trait was controlled by one dominant locus mapped to an interval of 92.0 kb on the Brassica A02 chromosome using SNP, SSR, and InDel markers. *BnaA02T0157000ZS* was identified as the candidate gene based on sequencing and gene expression analyses. Further analysis of individual plants revealed an adverse effect of up-curling leaf on photosynthetic efficiency. To conclude, the study provides a theoretical foundation for elucidating the leaf up-curling mechanism and breeding *B. napus*.

## Figures and Tables

**Figure 1 ijms-22-11693-f001:**
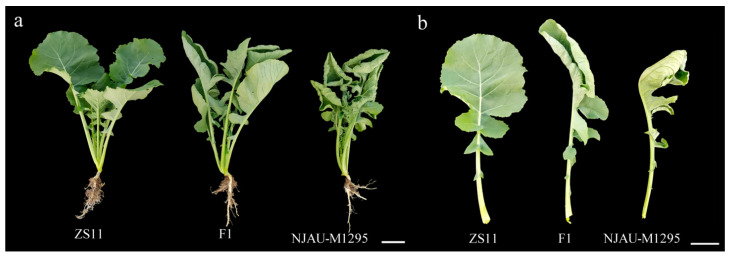
(**a**,**b**) Leaf phenotype at the seedling stage of ZS11, NJAU-M1295, and their F_1_ hybrid (Bar = 5 cm).

**Figure 2 ijms-22-11693-f002:**
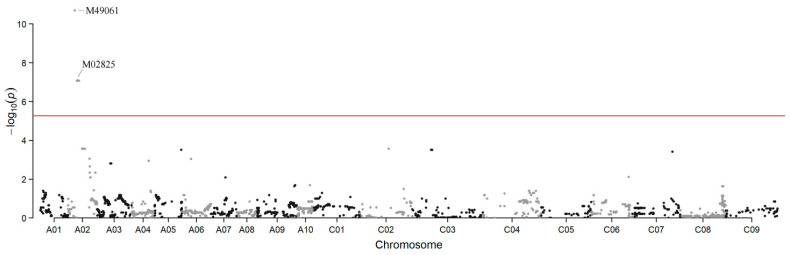
Manhattan plots of association analysis. Each dot represents an SNP. The horizontal red line represents the Bonferroni-corrected significance threshold −log10 (*p*) = 5.27.

**Figure 3 ijms-22-11693-f003:**
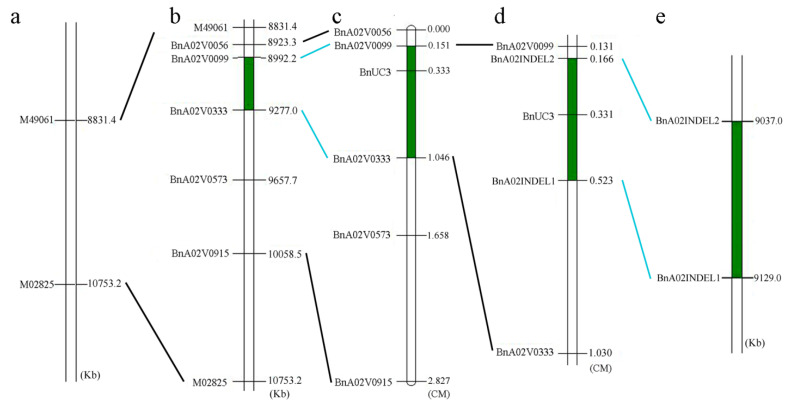
Mapping of the *BnUC3* locus (**a**–**e**). M49061 and M02825 are SNP markers on chromosome A02. Markers with prefix BnA02V are SSR markers. BnaA02INDEL1 and BnaA02INDEL2 are InDel markers. Physical map location is based on ZS11 reference genome sequence.

**Figure 4 ijms-22-11693-f004:**
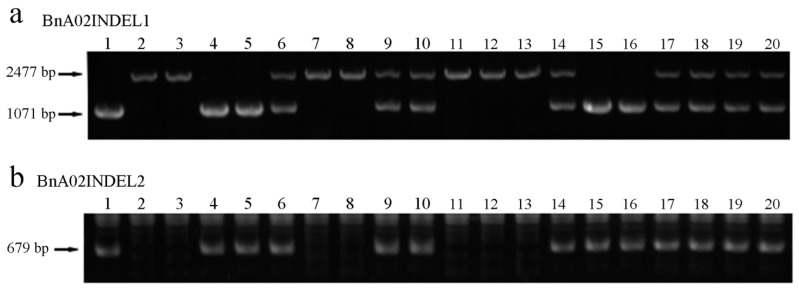
Partial molecular marker experimental results of the InDel mark BnaA02INDEL1 and BnaA02INDEL2. (**a**). The number 6, 9, 10, 14, 17, 18, 19, and 20 denote the PCR products from heterozygous plants with up-curling leaves, and 1, 4, 5, 15, and 16 denotes the PCR products from homozygous plants with up-curling leaves, and 2, 3, 7, 8, 11, 12, and 13 denote the PCR products from homozygous plants with flat leaves. (**b**). The number 1, 4, 5, 6, 9, 10, 14, 15, 16, 17, 18, 19, and 20 denote the PCR products from plants with up-curling leaves, and 2, 3, 7, 8, 11, 12, and 13 denote the PCR products from homozygous plants with flat leaves.

**Figure 5 ijms-22-11693-f005:**
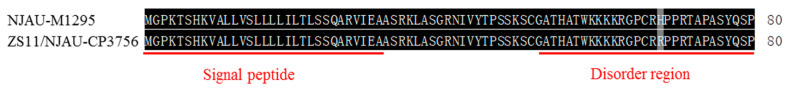
Amino acid sequence alignment of BnaA02T0157000ZS in NJAU-M1295, NJAU-CP3756, and ZS11.

**Figure 6 ijms-22-11693-f006:**
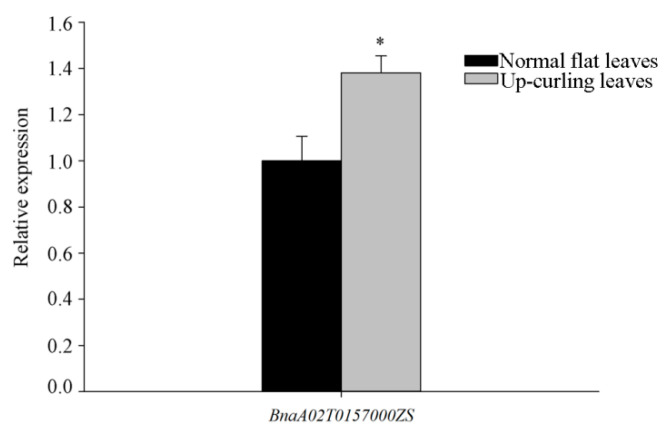
Gene expression analysis of *BnaA02T0157000ZS* by qRT-PCR in subpopulation with up-rolling leaves (gray) and the subpopulation with flat leaves (black)**.** Error bars represent the SE of three independent replicates. Statistical significance was calculated by two-tailed Student’s *t*-test: * *p* < 0.05.

**Table 1 ijms-22-11693-t001:** Leaf chlorophyll contents of homozygous up-curling leaf and flat leaf in F_2_ population derived from NJAU-M1295 and ZS11 (mean ± SD, n = 5 for each sample).

Genotye	Chl a (mg·g^−1^FW)	Chl b (mg·g^−1^FW)	Total Chl (mg·g^−1^FW)	Chl a/Chl b
flat leaf	1.27 ± 0.02	0.60 ± 0.03	1.86 ± 0.04	2.12 ± 0.08
up-curling leaf	1.26 ± 0.05	0.63 ± 0.03	1.90 ± 0.08	2.00 ± 0.04

**Table 2 ijms-22-11693-t002:** Leaf photosynthetic indicators of homozygous up-curling leaf and flat leaf in the F_2_ population (mean ± SD, n = 4 for each sample).

Genotye	Net Photosynthetic Rate (μmol CO_2_ m^−2^ s^−1^)	Stomatal Conductance (mol H_2_O m^−2^ s^−1^)	Intercellular CO_2_ Concentration (μmol CO_2_ mol^−1^)	Transpiration Rate (mmol H_2_O m^−2^ s^−1^)
flat leaf	20.86 ± 1.62	0.27 ± 0.04	236.25 ± 8.31	2.45 ± 0.16
up-curling leaf	12.25 ± 1.26 **	0.63 ± 0.17 **	341.85 ± 2.67 **	3.66 ± 0.36 **

Notes: ** Indicates significant differences between up-curling leaf and flat leaf plants at 0.01 level by *t*-test.

**Table 3 ijms-22-11693-t003:** Inheritance of *BnUC3* in populations derived from NJAU-M1295 vs. ZS11 and NJAU-CP3756 in *B. napus*.

	Population	Leaf Up-Curling	Normal Leaves	Expected Ratio	$\chi^{2}$	*p* Value
Cross I NJAU-M1295 vs ZS11	F_1_	87	0			
	RF_1_	77	0			
	F_2_	448	159	3:1	0.46	0.49
	BC_1_	86	92	1:1	0.20	0.65
cross II NJAU-M1295 vs NJAU-CP3756	F_1_	91	0			
	RF_1_	79	0			
	F_2_	498	158	3:1	0.29	0.58
	BC_1_	106	113	1:1	0.22	0.64

**Table 4 ijms-22-11693-t004:** Contingency table of the two SNP markers vs leaf up-curling trait in BC_1_ of ZS11×NJAU-M1295.

	M49061		M02825	
	**Aa**	**aa**	**Aa**	**aa**
Up-curling	33	1	32	2
Normal flat leaf	0	3	0	3
$\chi^{2}$	0.36		1.45	

**Table 5 ijms-22-11693-t005:** Function annotation of genes in the mapping interval.

Gene in *B.napus* cv. ‘ZS11’	Gene in *B.napus* cv. ‘Darmor’	Homolog in *Arabidopsis thaliana*	Putative Molecular Function
BnaA02T0156200ZS	BnaA02g12060	AT1G65295	Ubiquitin carboxyl-terminal hydrolase
BnaA02T0156300ZS	BnaA02g12070	AT1G65380	CLAVATA2|Leucine-rich repeat receptor-like protein
BnaA02T0156400ZS	BnaA02g12080	AT1G65410	NAP11 (NON-INTRINSIC ABC PROTEIN 11);
BnaA02T0156500ZS	BnaA02g12090	AT1G65420	Ycf20-like protein
BnaA02T0156600ZS	BnaA02g12100	AT1G65430	ARI8|Probable E3 ubiquitin-protein ligase ARI8
BnaA02T0156700ZS	BnaA02g12110	AT1G65440	GTB1|Transcription elongation factor SPT6 homolog
BnaA02T0156800ZS	BnaA02g12120	AT1G65450	GLC|HXXXD-type acyl-transferase family protein
BnaA02T0156900ZS	BnaA02g12130	AT1G65480	FT|PEBP family protein
BnaA02T0157000ZS		AT1G65510	STMP|secreted transmembrane peptides
BnaA02T0157100ZS	BnaA02g12140	AT1G65520	ECI1|Enoyl-CoA delta isomerase 1
BnaA02T0157200ZS	BnaA02g12150	AT1G65560	Zinc-binding dehydrogenase family protein

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
