# Peer review of "Identification and Fine Mapping of a Locus Related to Leaf Up-Curling Trait (Bnuc3) in Brassica napus"

_ijms, 2021, doi:10.3390/ijms222111693_

Round 1

Reviewer 1 Report

I found this manuscript particularly interesting. The authors identify and characterize some Brassica napus genes involved in leaf morphogenesis. The leaf is home to multiple biochemical processes underlying the life of the plant. In particular, the orientation of the leaf blade, the organization of the chloroplasts, the thickening of the cell wall are essential for the photosynthetic yield and therefore the life of the plant. These mechanisms are regulated by multiple genes that have been characterized in this manuscript.

In detail, the authors cross two varieties with different foliar phenotypes and photosynthetic yields, NJAU-M1295 with ZS11. In addition, they study the photosynthetic efficiency and the regulation of the expression of some genes, which are regulators of the morphological processes of the leaf, located in the BnUC3 locus.

The introduction is clear and well structured, the applied methods are sufficiently discussed, and the results obtained may have important applications in plant genetics for agronomic use.

However, I suggest the authors to better detail the statistical methods applied and to insert a recent manuscript on Brassica napus:

Zangani, E.; et al. Nitrogen and Phosphorus Addition to Soil Improves Seed Yield, Foliar Stomatal Conductance, and the Photosynthetic Response of Rapeseed (Brassica napus L.). Agriculture 2021, 11, 483. doi: 10.3390/agriculture11060483

Author Response

Response to Reviewer 1 Comments

Point 1: I suggest the authors to better detail the statistical methods applied and to insert a recent manuscript on Brassica napus:

Zangani, E.; et al. Nitrogen and Phosphorus Addition to Soil Improves Seed Yield, Foliar Stomatal Conductance, and the Photosynthetic Response of Rapeseed (Brassica napus L.). Agriculture 2021, 11, 483. doi: 10.3390/agriculture11060483

Response 1: Thanks you for your professional suggestions.

The statistical method for preliminary mapping the UC locus is a conventional linkage mapping method for qualitative trait locus for genome-wide scanning of target locus. We have explained in section 4.4. And we have added the statistical methods for sample comparison in section 4.7.

We have added these articles as reference. The reference papers suggested is appended and cited. This may be a help to discuss the relation between chlorophyll content and the crop yield.

Reviewer 2 Report

  1. No research hypothesis is presented in the introduction;
  2. The method of statistical evaluation of the survey data is not specified;
  3. In the Figure No. 6 is not explained, what the vertical bars indicate;
  4. I advise not to use very old literature sources.

Author Response

Response to Reviewer 2 Comments

Point 1: No research hypothesis is presented in the introduction;

Response 1: Thanks for your professional suggestions. We have provided the research hypothesis in the Introduction section.

Point 2: The method of statistical evaluation of the survey data is not specified;

Response 2: Thanks. The statistical method for preliminary mapping the UC locus is a conventional linkage mapping method for qualitative trait locus for genome-wide scanning of target locus. We have explained in section 4.4. And we have added the statistical methods for sample comparison in section 4.7.

Point 3: In the Figure No. 6 is not explained, what the vertical bars indicate;

Response 3: Thanks. Explanation of Figure No. 6 is modified as suggested.

Point 4: I advise not to use very old literature sources.

 Response 4: Thanks. We have added the some recent reports, and deleted some old literatures sun as References 6, 13, 17, 18, 23 and 24. Some old but classic literatures remain in our MS.